# Betanin purification from red beetroots and evaluation of its anti-oxidant and anti-inflammatory activity on LPS-activated microglial cells

Hosein Ahmadi[1,2,3], Zahra Nayeri[1,2], Zarrin Minuchehr[2], Farzaneh Sabouni[1]*, Mehdi Mohammadi[3]*

1 Department of Molecular Medicine, Institute of Medical Biotechnology, National Institute of Genetic Engineering and Biotechnology (NIGEB), Tehran, Iran, 2 Systems Biotechnology Department, Institute of Industrial and Environmental Biotechnology, National Institute of Genetic Engineering and Biotechnology (NIGEB), Tehran, Iran, 3 Bioprocess Engineering Department, Institute of Industrial and Environmental Biotechnology, National Institute of Genetic Engineering and Biotechnology (NIGEB), Tehran, Iran

* m.mohammadi@nigeb.ac.ir (MM); sabouni@nigeb.ac.ir (FS)

**Data Availability Statement:** All relevant data are within the paper.

## Abstract

Microglial activation can release free radicals and various pro-inflammatory cytokines, which implicates the progress of a neurodegenerative disease. Therefore suppression of microglial activation can be an appropriate strategy for combating neurodegenerative diseases. Betanin is a red food dye that acts as free radical scavenger and can be a promising candidate for this purpose. In this study, purification of betanin from red beetroots was carried out by normal phase colum chromatography, yielding 500 mg of betanin from 100 g of red beetroot. The purified betanin was evaluated by TLC, UV-visible, HPLC, ESI-MASS, FT-IR spectroscopy. Investigation on the inhibitory effect of betanin on activated microglia was performed using primary microglial culture. The results showed that betanin significantly inhibited lipopolysaccharide induced microglial function including the production of nitric oxide free radicals, reactive oxygen species, tumor necrosis factor-alpha (TNF-α), interleukin-6 (IL-6) and interleukin-1 beta (IL-1β). Moreover, betanin modulated mitochondrial membrane potential, lysosomal membrane permeabilization and adenosine triphosphate. We further investigated the interaction of betanin with TNF-α, IL-6 and Nitric oxide synthase (iNOS or NOS2) using in silico molecular docking analysis. The docking results demonstrated that betanin have significant negative binding energy against active sites of TNF-α, IL-6 and iNOS.

## Introduction

Increasing evidences demonstrate that microglial activation and inflammatory response pathway may play crucial role in the pathogenesis of neurodegenerative diseases [1] Microglia are the most important immune cells of the central nerves system (CNS). These macrophage-like

**Funding:** This work was financially supported by the National Institute of Genetic Engineering and Biotechnology (grant number: 595) to FS.

**Competing interests:** The authors have declared that no competing interests exist.

cells, emerge early erythro-myeloid pre-cursors in the embryonic yolk sac and migrate to the CNS before the formation of macrophages from hematopoietic stem cells (HSCs) [2]. Commonly, highly ramified cells are determined as 'resting' form of microglia with proliferative functions while the amoeboid cells are defined as 'activated' form. The activated form possess inflammatory functions which is raised from the presence of pro-inflammatory cytokines and free radicals of nitrogen and oxygen [3]. Studies have shown that activation of microglia can be regarded as an indicator for neuroinflammation and neurodegenerative diseases [4]. When microglia gets activated, pro-inflammatory cytokines such as IL-1β, IL-6, TNF-α and NO$^•$ secreted through NF-κB transcription factor that has been activated by Toll-like receptors (TLRs) and eventually involved further damage of neurons [5]. Microglia expresses specialized pattern recognition receptors (PRRs) in the CNS that can initiate neuroinflammation. These PRRs are capable of stimulating the microbial molecules which are known as pathogen-associated molecular patterns (PAMPs) [6]. TLRs, including TLR2 and TLR4, are major PRRs that can be stimulated by these substances and initiate an inflammatory response [7]. Thus, managing microglial activation, neuroinflammation, and oxidative stress raised by reducing free radicals may demonstrate therapeutic benefits in neurodegenerative diseases.

In healthy cells, there is a direct relation between free radical and endogenous antioxidant defense processes [8]. However, in degenerative diseases such as neurodegenerative disorders and neuroinflammation, this equilibrium has been destroyed [9]. Therefore, this oxidative stress can hurt vital macromolecules such as membrane lipids, proteins, and DNA which can probably result in cell death. Hence antioxidant materials may have beneficial effects on neurodegenerative disease by reducing free radicals and suppressing oxidative stress [8,10].

Many studies have proved that plants have different biological activities [11,12]. Betalains are one of the pivotal plant pigment families that are mostly found in Amaranth [13]. Betalains are divided into two subgroups of betacyanins with red-violet and betaxanthins with yellow-orange color. Both of these groups are water-soluble nitrogen-containing pigments and have free radical scavenger activity [11,13]. Betanin with red color is widely found in red beet and is the most common betacyanin pigment that acts as a stimulator of antioxidant defense mechanisms and has a considerable free radical scavenger activity [11,13]. Several investigations have also reported beneficial impact of betanin as anti-inflammatory factor [14]. The ability of betanin for suppression of cancer cells, lipid peroxidation and heme disintegration in in-vitro has been specified [15–17].Furthermore, a most recent report exhibited that betanin significantly suppresses NF-κB DNA-binding activity in rats stimulated with acute renal harm [18]. In another study Reddy et al., have shown that betanin suppresses 97% of enzyme activity in cyclooxygenase-2 (COX-2) that was comparable or even greater than some phenolic compounds and several anti-inflammatory drugs like Celebrex, Vioxx, and Ibuprofen [19].

In fact the anti-inflammatory processes of natural compounds have been demonstrated in numerous investigations and have been proved in innumerable preclinical studies [10]. Furthermore the excess costs and side effects of using nonsteroidal anti-inflammatory drugs, turned natural anti-inflammatory elements into more popular compounds [20–22]. Patel NK et al. showed that ethyl acetate extraction of *Cassia occidentalis* roots, suppresses LPS-induced IL-1β, TNF-α, and NO$^•$ release in macrophages. The anti-inflammatory effects of this extract have been determined to be equally potential both in *in vitro* and *in vivo* models [23]. Despite the proved significant antioxidant properties of betanin, there is no information available on its anti-neuroinflammatory effects. In this study, we report the results of investigation on the anti-inflammatory effect of betanin in stimulated microglial cells for possible effects on reducing neuroinflammation.

## Materials and methods

### Materials

The used red beetroots were from Dezful (in the south of Iran) and purchased from the local market in the city of Dezful, Khoozestan, Iran. PBS (Phosphate buffered saline), DMEM (Dulbecco's Modified Eagle Medium) medium, FBS (fetal bovine serum) were purchased from GIBCO, UK. MTT [3-(4,5-dimethylthiazol-2-yl)-2,5-diphenyl tetrazolium bromide], LPS (lipopolysaccharide), Betanin (CAS 7659-95-2), sulphanilamide, TLC (Thin-layer chromatography sheets), silica gel (high-purity grade pore size 60 Å, 60–100 mesh) and NED (N-1-naphthyl ethylenediamine dihydrochloride) were from Sigma-Aldrich, USA. methanol, and acetonitrile were HPLC-grade, citric acid, ascorbic acid, phosphoric acid and DMSO (Dimethyl sulfoxide) were obtained from Merck, Germany. https://www.biocompare.com. Anti-CD11b antibody (ab128797, Cambridge, UK, https://www.abcam.com), Alexa-488 Cross-adsorbed anti-rabbit IgG (H+L) (A11008, Invitrogen, https://www.thermofisher.com).

### Instrumentation

Flow cytometer (FACS Calibur, BD Bioscience, USA). Mass Spectrometer Agilent Technologies 5975C, HPLC (High-Performance Liquid Chromatography) Agilent Technologies 1260 Infinity II LC System with Eurospher 100–5 C18 with precolumn, Column 250 x 4.6 mm (KNAUER *25VE181ESJ*), UV-Vis Microplate Spectrophotometer (Epoch 2 BioTek), IR Spectrophotometer (Bruker Tensor 27)

### Methods

The authors confirm that all experiments were performed in accordance with relevant guidelines and regulations. The study was also reviewed and approved by the Bioethics Committee of the Health Ministry (Tehran, Iran; permit no. IR.NIGEB.EC.1395.4.1.C).

**Betanin extraction and purification.** Chopped red beetroots were extracted into methanol/water (80/20 v/v) the solvent with 50 mM ascorbic acid at a solid/liquid ratio of 1/5 (g/mL) for 30 minutes under continuous mechanical stirring, nitrogen bubbling and light protection. The pH of the extraction solvent was adjusted to 5.5 for more stability of betalains. The supernatant was centrifuged at 15000 g for 30 minutes at 4°C to remove the solid components. Column chromatography was then used for purification of betanin from the extract with the elution solvent mixture of methanol/water/glacial acetic acid (9/0.3/0.7). Purification of betanin was monitored continuously by TLC analysis and the pure fractions were concentrated under vacuum at 35°C and freeze-dried. The prepared betanin was identified and characterized by using TLC, UV-visible, HPLC, FT-IR, ESI-MASS spectroscopy.

**Chromatography (RP-HPLC).** HPLC analysis was performed with a C18 reversed-phase column with a particle size of 5μm in a gradient of elution including two solutions. Solution A contains 100% water and 0.1% (v/v) TFA and solution B contains 100% acetonitrile. The gradient was carried out for 40 min at a flow rate of 0.5 ml/min. The elution profile was 0–4 min 100% A, 4–20 min 0–14% B in A (linear gradient), 20–25 min 14% B, 25–30 min 14–50% B, 30–34 min 50–100% B, 34–40 min 100% B. The detection wavelengths was adjusted to 538 nm (betanin $\lambda_{max}$).

**Antioxidant activity.** Antioxidant activity was assayed by DPPH. To this, 250 μM of DPPH solution in methanol (80%) was added to sample solutions containing 10 μM of pure betanin, standard betanin, and ascorbic acid in deionized water. After 30 minutes of incubation, the absorbance was detected using a spectrophotometer at 518 nm.

**Primary microglial cell culture and treatment.** Microglia primary culture was obtained from the cerebral cortex brains of 1–3 days old Wistar rat. Concisely, the mix glial cells were cultured for two weeks in DMEM high glucose, supplemented with 10% FBS. Microglial cells were isolated with a Shaker at 80 rpm for 30 min at 37˚C. Harvested microglial cells were placed into 96-well plates at a density of $3 \times 10^4$ cells/well [24]. To confirm the purity of the harvested microglial cells, anti-CD11b antibody (ab128797, 1:200; Abcam), a typical microglia marker was used on FACS Calibur flow cytometer instrument and the secondary antibody was used Alexa-488 conjugated anti-rabbit IgG (A11008, Invitrogen).

Treatment of microglia cells was performed in three timepoints 24, 48 and 72h; each timepoint consist of the control group, without LPS and betanin, induced group with 1μg/ml LPS [24] and a group with 5 different doses of betanin, contains 100, 200, 300, 400 and 500μM which were selected for viability assay.

**Cell viability assay.** Cell viability was performed by the MTT assay. 20 μl of MTT solution (5 mg/ml) was added to each well with a total volume of 200 μl culture medium and incubated at 37˚C for 4h. Afterward the medium was removed, and 100 μl of DMSO was added. The absorbance was detected using a microplate spectrophotometer at 580 nm after 30 minutes.

**Nitrite assay.** Aggregation of nitrite levels was evaluated in the microglia culture medium after 24h of stimulation by the Griess reaction. Nitrite was measured as a criterion of NO• production. 50 μl of the culture medium was blended with an equal volume of 1% sulphanilamide in 5% phosphoric acid in a 96-well plate. The above combination incubated at room temperature for 10 minutes, and then 50 μl of 0.1% NED was added to the combination, and after 10 minutes of incubation, the absorbance value was detected using a microplate spectrophotometer at 540 nm [3].Dilution of Sodium nitrite was performed in culture medium at various concentrations, from 0 to 100 mM, to obtain a standard curve.

**Extracellular cytokines assay.** Microglia cells ($5 \times 10^5$ cells/well in a 6-well plate) were treated with LPS in the presence or absence of betanin. The culture medium of the microglia cells was gathered in 24 hours after effective material stimulation. The levels of inflammatory cytokines comprising TNFα, il-1β, and IL-6 in culture medium were assessed using enzyme-linked immunosorbent assay kits conforming to the constructor's instructions. The absorbance value was detected using a microplate spectrophotometer at 450 nm.

**Real-time PCR.** Cells were lysed in RNXplus Reagent (Sinaclon, Iran), and the total RNA was extracted according to the manufacturer's instructions. The quality, quantity, and concentration of extracted RNA were evaluated by electrophoresis and Nano-Drop. RNA was reverse transcribed to cDNA by Synthesis Kit (Thermo Scientific, USA). Quantitative PCR was executed using the MIC qPCR–Magnetic Induction Cycler and RT-PCR kit with SYBR Green (Ampliqon, UK) according to the manufacturer's instructions. The condition of PCR was as follows: 5 min enzyme activation at 95˚C, 40 cycles of 95˚C for 30 s, annealing temperature of primers for 20 s and 72˚C for 40 s, so for *Tnf-α*, the forward primer: 5′GCTCCCTCTCATCAGTTCC3′ and the reverse primer: 5′TTGGTGGTTTGCTACGAC3′, annealing Tm 55˚C; for *Nos2*, the forward primer: 5′GAGATGTTGAACTACGTCCTATC-3′, the reverse primer: 5′CCATGACCTTCCGCATTAG-3′, annealing Tm 60˚C and for *Nf-κb*, the forward primer: 5′GCTCAAGATCTGCCGAGTAAA-3′ and the revers primer: 5′GTCCCGTGAAATACACCTCAA-3′, annealing Tm 62˚C.Moreover, a reference gene *Gapdh* was amplified by the forward primer: 5′CTCATGACCACAGTCCATGC3′ and the reverse primer: 5′TTCAGCTCTGGGATGACCT3′.

**Lysosomal membrane stability assay.** Lysosomal membrane stability of microglial cells was specified with the redistribution of acridine orange as a fluorescent dye. Acridine orange (AO) is a lysosomotropic weak base fluorescent dye that its fluorescence emission depends on concentration. In intact lysosomes, due to a low pH inside AO accumulates and emits red

fluorescence, while in the cytosol and the nucleus, it emits green fluorescence. When the lysosome ruptures, the AO leaks out and redistributes in the cytoplasm. Hence, AO redistribution can be used to determine the lysosomal membrane stability [25,26]. Microglia cells were stained with acridine orange (5 μM) and the excess fluorescent dye were washed off by performing centrifugation twice at 1500 rpm for 3 min in the 2ml incubation medium. Acridine orange redistribution in the microglia cells was measured by a fluorescence spectrophotometer at 490 nm for excitation and 540 nm for emission wavelengths.

**Determination of adenosine triphosphate concentration.** Adenosine triphosphate (ATP) concentration in mitochondria was measured by a bioluminescent somatic cell assay kit (Sigma Aldrich.MO 63103, USA). The intensity of bioluminescence was determined by a Sirius tube luminometer (Berthold Detection System, Germany).

**Determination of mitochondrial membrane permeability and reactive oxygen species.** Microglia cells ($5 \times 10^5$ cells) were treated with betanin/LPS for 24 h. After treatment, microglia cells were washed with PBS. H2DCFD and Rhodamin 123 (10 mM) were applied to determine intracellular reactive oxygen species (ROS) and disrupted cell membrane. Deesterification is the reason for diffusing these factors into the cells. Following reactions with peroxides generate fluorescent 5-chloromethyl-2′, 7′ dichlorofluorescein (DCF). Mitochondrial membrane permeability (MMP) was specified using lipophilic cationic dependent fluorescent dye rhodamine (Rh123) by flow cytometry.

Cells were scanned on a FACS Calibur flow cytometer instrument to determine the mitochondrial membrane permeability (MMP), and light scattering was analyzed for 10000 counts per sample. Using Flowing software (ver-2-5-1), argon-ion laser sets as 488 nm and fluorescence signals were measured in a 530 nm (FL-1 channel).

## In silico analysis

In order to predict the inhibitory effect of betanin on IL-6, TNF-α, iNOS and NF-κB, in silico approach (blind and accurate docking) was employed. In the first step, Protein modeling for preparation of IL-6, TNF-α, iNOS and NF-κB, was performed. For this, I-TASSER (https://zhanglab.ccmb.med.umich.edu/I-TASSER/) [27] RaptorX (http://raptorx.uchicago.edu/) [28] and SWISS-MODEL (https://swissmodel.expasy.org/) [29] were used to predict structures. Also for the protein structure validation, the "SAVES v5.0" server (http://servicesn.mbi.ucla.edu/SAVES/) was applied, and the Ramachandran plot for each structure was obtained. To generate improved 3D models, Molecular dynamics (MD) simulation, was applied using GROMACS package 5.1 by 54A7 force field and the simple point charge (SPC) water model was used to develop the solvated systems. Afterward, to neutralize the systems for each simulation, appropriate ions were added. Energy minimization was achieved, and system equilibration was performed under NVT and NPT ensembles for 100 ps. MD simulation was performed for a time duration of 50 ns at 300 K temperature and 1 bar pressure. Finally, the root mean square deviation (RMSD) and root mean square fluctuation (RMSF) was calculated using the GROMACS 5.0 "rmsdist" algorithm and "rmsf" algorithm, respectively. On the other hand, for the preparation of ligand, betanin structure was retrieved from the PubChem database (https://pubchem.ncbi.nlm.nih.gov/). Then, to acquire the optimized geometry of the neobaicalein structure, the webserver ATB, (Automated force field Topology Builder, http://compbio.biosci.uq.edu.au/atb), was used. Accordingly blind docking studies were carried out. Eventually accurate docking on this binding pockets was performed by AutoDock Tools (ADT) by AutoDock Vina 1.1.2.

**Statistical analysis.** All results were determined as mean ±S.D of at least three independent experiments. Statistical analysis of data was done by one-way ANOVA using the Holm-

Sidak method in multiple comparisons of means by Sigma Plot Version 12.0. In the RT-PCR, Data were analyzed using the LinRegPCR, and REST 2009 software also results were shown as fold differences.

## Results and discussion

### Purification and characterization of betanin

Purification of betanin from the extract of red beetroot was performed by using normal phase column chromatography (NPC). The used purification methodology yielded 500±22 mg of betanin from 100 g of red beetroot. The efficiency of purification is about five times higher than the maximum value that has already been reported in other studies. This is more likely because of high amounts of betanin in the used source of red beetroot. Several studies have reported quantification of different amount of betanin in red beet root ranging from 2.8–8.5 g/kg [30]. Kanner et al. applied HPLC with a C18 column to purify betanin, obtaining 60 mg of pure betanin from 100 g red beet [18]. Sephadex gel column has also been evaluated for the purification of betanin in another study, yielding 15–50 mg betanin from the same amount of beetroot extract [31]. Comparison of seven different methods for purification of betanin also showed that the use of NPC and ion exchange chromatography resulted in purification of 31.4 mg and 89.1mg of betanin from 100 g of fresh beetroot, respectively [32]. The purified betanin was evaluated by using several methods and comparing the obtained results with those of the standard sample of betanin from commercial source (Fig 1).

The TLC analysis (Fig 1i) and IR spectroscopy (Fig 1iii) of the purified betanin showed almost a same pattern and retention factor with that of the standard sample. UV spectroscopy was also used to characterize the purified sample. As can be seen from Fig 1ii both the purified sample and also the control show a single peak at 538 nm which belongs to λmax of betanin, while the crude extract shows two peaks at 450 nm and 538 nm. Elimination of the signal at 450 nm of the crude extract together with identical pattern of the commercial and purified spectra clearly confirms the successful purification of betanin. Also, ESI-mass was used for determination of purified betanin molar mass (Fig 1v). The mass calculated by ESI-mass for the purified sample was about 552 dalton which is identical with the molar mass of betanin. HPLC analysis for the crude extract, purified, and standard betanin were performed at 538 nm corresponding to the maximum absorption of betanin (Fig 1iv). These results indicated the relative purity of betanin as compared to the standard sample. This result further confirmed that the purification procedure was effective in isolating betanin in its purified form. Evaluation of antioxidant activity using DPPH for the purified and standard samples is shown in Fig 1vi. DPPH is a free radical, which produces a violet solution in methanol (80%) that is decreased in the presence of a free radical scavenger like natural antioxidants. The results showed almost same antioxidant activity for the commercial and purified samples at the same concentration (Fig 1vi). However ascorbic acid as positive control has more activity compared to the betanin.

### Cell isolation and cell viability

Immunofluorescence study with flow cytometry analysis of CD-11b antibody for performing harvested microglial cells showed more than 94% CD-11b+ cells. This means that microglia cells are the dominant population of the total isolated cells (Fig 2i). Assessment of cytotoxicity for purified betanin on microglial cells at 24, 48, and 72 hours using MTT assay was performed. According to Fig 2ii, for the highest concentration tested (500 μM) no toxicity was observed. Drug toxicity is known to be a drug's weaknesses. The lack of toxicity even at high concentrations can suggest the probable potential of betanin to be used as a new strategy for anti- inflammatory treatment.

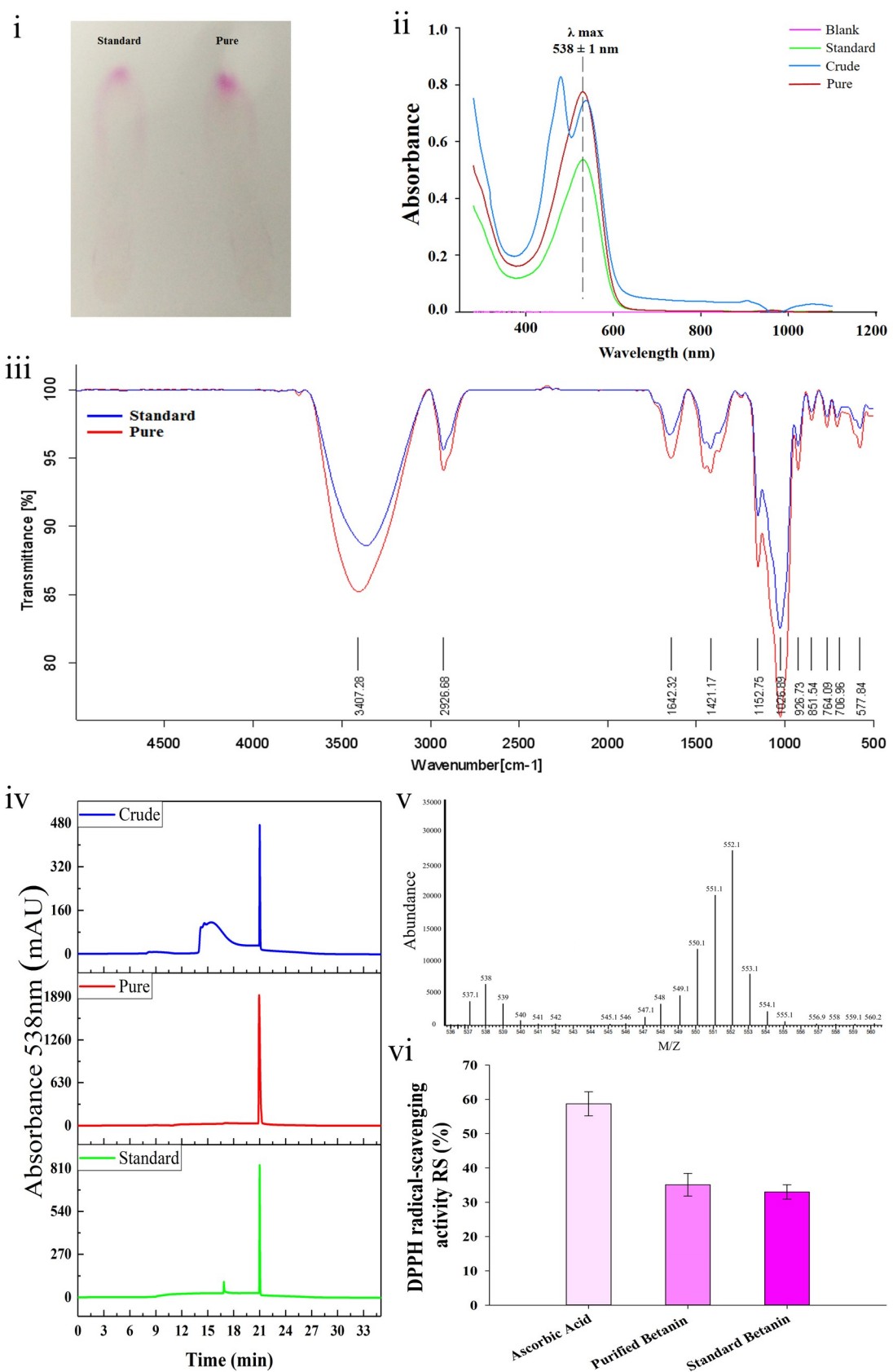

**Fig 1. Purification and characterization of betanin.** Evaluation of purified betanin. TLC (i), Optical absorption (ii), and FT-IR (iii). Comparison of purified and standard betanin. HPLC chromatogram (iv), Mass spectrum (v), Antioxidant activity using DPPH assay (vi).

## Anti-inflammatory effects of betanin in LPS- induced microglial cells

Neuroinflammation is one of the most critical mechanisms of neurodegenerative diseases. It can also lead to the production, and release of free radicals [33]. Microglial cells are one of the most important cells involved in neuroinflammation and they play a key role in free radical generation [3]. The production of NO$^•$ is mediated by the iNOS protein in microglial cells, which could predispose the oxidative stress and ROS formation [34]. Therefore, inhibition of free radicals, especially RNS, is one of the main ways to combat neuroinflammation. Application of antioxidants has been suggested as a way to reduce the amount of these free radicals [35]. In order to examine the anti-inflammatory properties of betanin as a natural antioxidant, we first investigated its effect on nitrogen and oxygen free radicals in LPS-induced microglial cells, followed by the evaluation of MMP, ATP, redistribution of lysosome and pro-inflammatory cytokines analysis.

## Investigation of NO• level as an inflammatory marker

As can be seen from Fig 2iii, morphological transformation of microglia from ramified (Fig 2iiia) into amoeboid occurs (Fig 2iiib). A five-fold increase of NO$^•$ concentration (Fig 2iv) in LPS-induced cells confirms the inflammation model. The pretreatment of betanin at 100 and 200 μM concentrations was not effective in inhibiting NO$^•$. However, in the betanin concentrations of 300 and 400 μM, there was a slight decrease in NO$^•$ concentration compared to the LPS-induced group. With increasing betanin concentration to 500 μM, the lowest NO$^•$ level was observed as compared to the LPS-induced group, still almost 1.5 fold higher than the NO$^•$ concentration of the control group. Due to the high cell death in neuroinflammation process, the non-toxicity of betanin even at high doses has great importance for the use as an anti-inflammatory agent. Therefore, according the above mentioned observations, the concentration of 500 μM was chosen for the rest of analysis. NO• as a crucial factor in neuroinflammation, has some degenerative effect on the brain. It can cause excitotoxicity due to stimulating astrocytes for release glutamate as well as NO• inhibit the mitochondrial respiratory chain and induce production of peroxynitrite (ONOO$^−$) as a fatal free radical [34].

## ROS, MMP, ATP, and redistribution of lysosomes analysis

ROS was quantified by flow cytometry analysis (Fig 3i). As expected, the ROS level in the LPS-induced group was significantly increased compared to the control group. Pretreatment of the LPS-induced cells with betanin causes a significant reduction in ROS level while there was no meaningful difference between pretreatment and control group. Bearing in mind the correlation between ROS production with oxidative stress and mitochondrial dysfunction [36]. One of the vital functions of mitochondrial is to maintain the MMP. We monitored the changes in MMP to find more information about the effect of betanin in mitochondrial dysfunction with Rh123 as a sensitive cationic fluorescent probe for membrane potential (Fig 3ii).

The results show a significant peak shifting of MMP in the LPS-induced group as compared to the control group, which demonstrates that the mitochondrial membrane potential is impaired. However, after pretreatment, betanin can decline mitochondrial membrane potential compared to the LPS-induced group (Fig 3ii). Collapse in MMP is related to the induction of inflammation signaling and oxidative stress. It is well documented that the mitochondrial

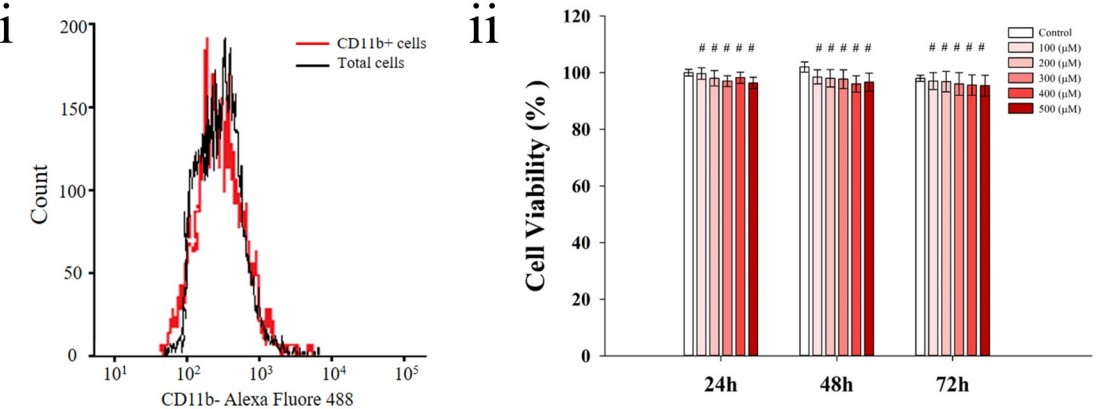

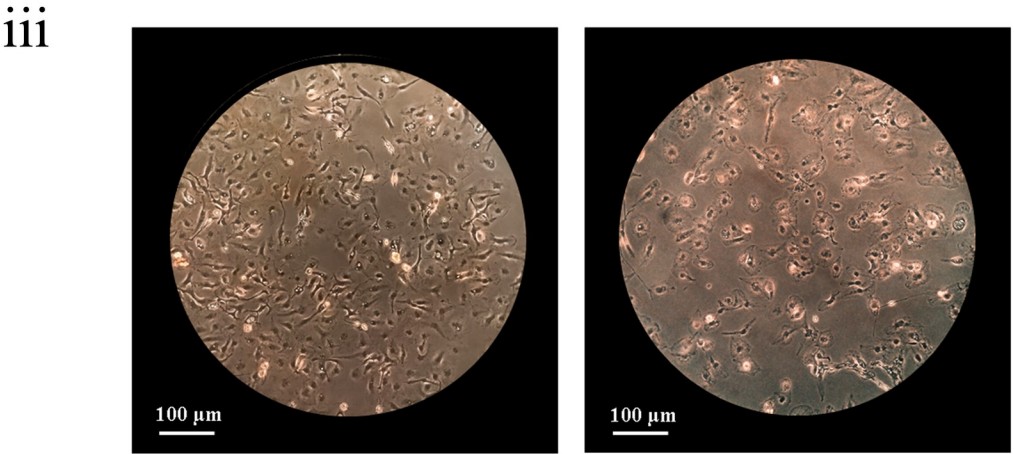

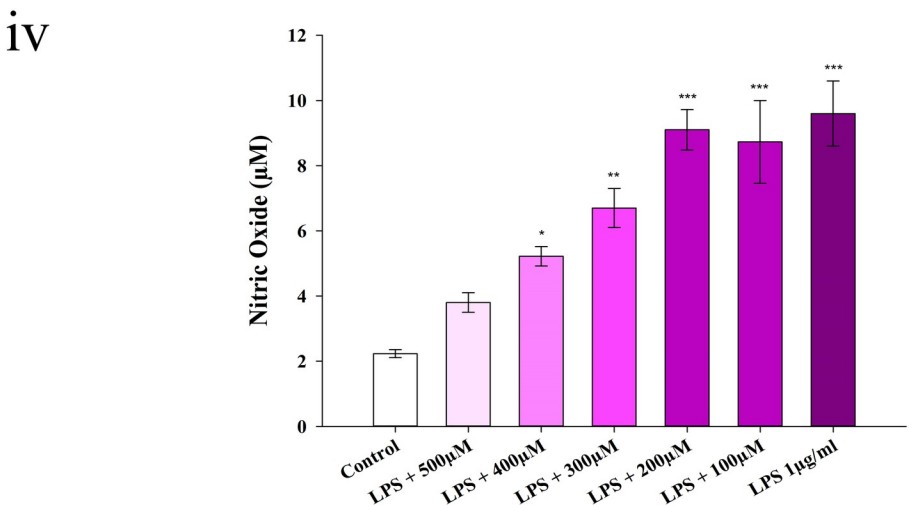

**Fig 2. Cell isolation, cell viability and investigation of NO• level.** Flow cytometry analysis of CD-11b+ cells (Fig i). Cytotoxicity of betanin on microglial cells using the MTT assay (Fig ii). The control group is the cells treated with PBS as betanin solvent. Evaluation of LPS-induced microglial cells. Morphology characteristics of microglia cells before and after LPS induction with 200x magnification light microscope (Fig iii). Before induction, cells are healthy and ramified (a), after LPS-induction, cells switch to the inflamed and amoeboid form (b). Determination of nitric oxide by Griess reaction (iv). All groups were compared with the control group. The difference in NO• concentration in the LPS group and control group was more than five times. However, no significant difference was detected between the control and 500 μM betanin. *P<0.05, **P<0.01, ***P<0.001 and #P>0.05 or non-significant, one-way ANOVA.

dysfunction and oxidative stress are correlated with the amount of ATP production in cell [37]. Therefore, for further investigation on mitochondrial activity, ATP levels were measured in all three groups (Fig 3iii). The results in Fig 3iii show a significant depletion of ATP level in the LPS-induced cells compared to the control group. Although betanin pretreatment leads to an increase in ATP levels as compared to the LPS-induced group, its level changes indicate a positive trend in mitochondrial dysfunction and oxidative stress inhibition. Oxidative stress and mitochondrial dysfunction also affect lysosomal membrane stability. Increased lysosomal membrane permeabilization causes the leakage large quantities of lysosomal materials such as acidic hydrolases to the cytoplasm and further mitochondrial damage and promotes the irreversible degradation of lipids and proteins, thus intensifying the inflammation [38,39]. Therefore, the LMP was evaluated in all three groups. As shown in Fig 3iv, a two-fold increment of the LMP was observed for LPS-induced group. There was also a significant difference between the pretreated and LPS-induced groups. Consequently it can be concluded that pretreated cells with betanin will have less inflammation than the LPS-induced group.

## Evaluation of pro-inflammatory cytokines

Considering the role of cytokines in neuroinflammation and the involvement of the immune system in this complex process, the evaluation of inflammatory factors such as TNF-α, IL-1β, and IL-6 is of great importance [5]. The increase of free radicals, oxidative stress, mitochondrial dysfunction, lysosomal membrane permeability, and critical inflammatory cytokines indicate that M1 state in microglia is occurred [40]. In this study, three important IL-6, IL-1β, and

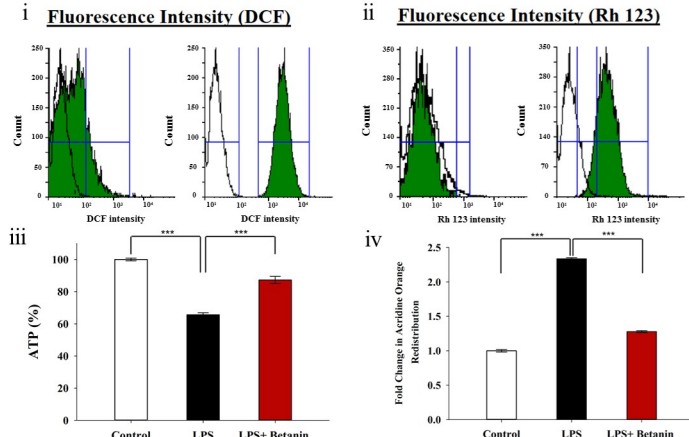

**Fig 3. ROS, MMP, ATP, and redistribution of lysosomes analysis.** Flow cytometry, ATP, and LMP analysis of microglial cells before, and after pretreatment with betanin at 500 μM concentration. Flow cytometry histogram of ROS (i) and MMP (ii), comparison between control and pretreated (left), control and LPS-induced groups (right). ATP levels (iii). LMP analysis by acridine orange redistribution. The absorbance of the control group was considered as one (iv). *P<0.05, **P<0.01, ***P<0.001, one-way ANOVA.

TNF-α cytokines were investigated (Fig 4). As expected, a significant increase in the LPS-induced group was observed. These results of inflammatory cytokines confirmed that the inflammation occurs which is usually associated with M1 phenotype. The most significant difference of cytokines between the control and LPS-induced groups belongs to TNF-α, indicating the critical function of this cytokine in inflammatory model. Surprisingly, all three cytokines were declined in the pretreated group. It shows that betanin has a positive effect on inhibition of these cytokines. Switching of the microglia phase from M1 to M2 can reduce inflammatory activity and improve its efficiency. In the M2 state, inflammatory cytokines, RNS, and ROS decrease, while mitochondrial activity increases [40,41]. Therefore, conversion from M1 to M2 requires improvement of mitochondrial activity and reduction of free radicals and inflammatory cytokines, while mitochondrial dysfunction can prevent the conversion [42]. The results demonstrated that betanin is capable to play these roles and also helps to reprogramming of microglial cells.

## Comparison of *Nos2, Tnf-α* and *Nf-κb* expression

NO• plays an *important role* in neuroinflammation, mitochondrial dysfunction, and microglial reprogramming from M1 to M2 [41,42]. The effect of NO• on TNF-α and their cross-talk intensify the LMP degradation [38,43]. Also, we already showed the high inhibitory effect of betanin on NO• concentration. Hence, the expression of *Nos2* was investigated. The real-time results showed a significant increase in *Nos2* expression in the LPS-induced group, proving microglial inflammatory activity. Furthermore, a significant decrease in the expression of this gene was observed in pretreated microglial cells compared to LPS-induced cells (Fig 5). Our finding reveals the dual role of betanin in both antioxidant activity and *Nos2* reduction expression level. According to the NO• and TNF-α cross-talk and our product analysis, the positive effect of betanin on reducing expression of *Tnf-α* would also be predictable. As shown in Fig 5, there was a significant increase in *Tnf-α*, expression in the LPS- induced group. The pretreated group with betanin demonstrated a significant decrease in *Tnf-α* expression. Other studies have reported the inhibition of this cytokine by antioxidants [44]. TNF-α leads to the expression of pro adhesive molecules in endothelial cells that gives rise to the penetration and accumulation of leukocytes in the brain, and eventually makes more extensive neuroinflammation [45]. The binding of TNF-α to its receptor activates other glia cells, gliosis and enhances ROS production [43,45]. Activation of the p38, JNK, and NF-κB pathways by the TNFR1 (Tumor necrosis factor receptor 1) enhances the expression of *Nos2* and production of NO•. The effects of TNF-α on ROS and RNS stimulates these radicals to increase TNF-α self-expression and produces a positive feedback loop which consequently enhances the neuroinflammation [43]. In LPS-activated macrophages, p38 immediately phosphorylated at a tyrosine residue and performs a crucial role in inflammation [46]. Activating of p38 MAPKs is involved in the upregulation of inflammatory mediators, including TNF-α, iNOS, and cyclooxygenase-2 (COX-2) [47]. The obtained results of attenuating in the expression of Tnf-α and Nos2 at mRNA level, TNF-α protein, and the product of iNOS (NO•), could be attributed to either the direct effect of p38 on these factors or the negative effect of betanin on the performance of p38. Another important protein that is involved in inflammation is NF-κB. This transcription factor was also proved for its contribution in the expression of Tnf-*α* and *Nos2* genes by regulating their expression. Furthermore, NF-κB has been associated with inflammatory diseases [48]. The activation of NF-κB affects phosphorylation of p65, ubiquitination, phosphorylation and degradation of IκBα which intensifies the translocation of NF-κB to the nucleus thus enhancing the expression of its downstream genes [48,49]. Therefore, one of the possible reasons for the decrease in *Tnf-α* and *Nos2* genes expression could be the effect of betanin on the expression

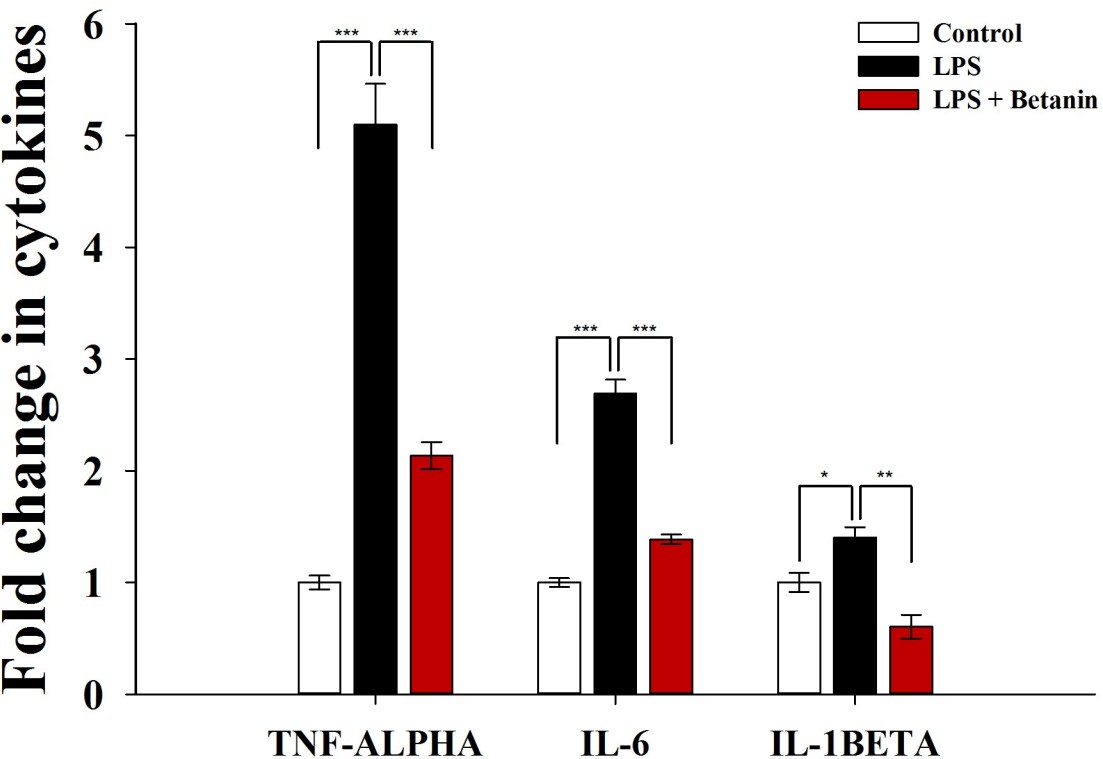

**Fig 4. Evaluation of pro-inflammatory cytokines.** Comparison of cytokines levels in microglial cells after pretreatment with 500 μM of betanin. Concentration of TNF-alpha: 82, 418, 75 pg/ml, IL-6: 261, 705, 363 pg/ml and IL-1BETA: 2573, 3610, 1553 pg/ml were obtained for control, LPS and LPS+Betanin groups, respectively. The experiment was performed with three replications at two different times. $^{*}$P<0.05, $^{**}$P<0.01, $^{***}$P<0.001, one-way ANOVA.

of the *Nf-κb* gene and function of this transcription factor. The quantitative polymerase chain reaction (qPCR) results exhibit a significant increase in *Nf-κb* at LPS-induced group that can be a potential reason for inflammatory signaling pathways. Also at the pretreated group with betanin, a significant reduction in the expression of the *Nf-κb* gene was observed. In the inflammation, JNK-AP-1 and IKK-NF-kB signaling cascades can amplify the production of several pro-inflammatory cytokines such as TNF-α, IL-6, iNOS and IL-1β [50,51].

These data suggest that betanin probably is able to inhibit *Tnf-α* and *Nos2* genes and their transcription factor (*Nf-κb*). Accordingly betanin might play a crucial role in neuroinflammation inhibition by the loop dysfunction that further confirms the potential of betanin to be used as a new natural anti-inflammatory agent.

## Protein modeling and docking study

Further investigation of the anti-inflammatory potential of betanin was performed by molecular docking simulation on Pro-inflammatory cytokines (IL-6 and TNF-α). According to the role of betanin in reducing NO$^{•}$ and mRNA level of NOS2, the inhibitory potential of betanin on the iNOS enzyme was studied. On the other hand, LPS induced activation of the microglial cells via the NF-κB signaling pathway. The activation of NF-κB-p65 releases inflammatory

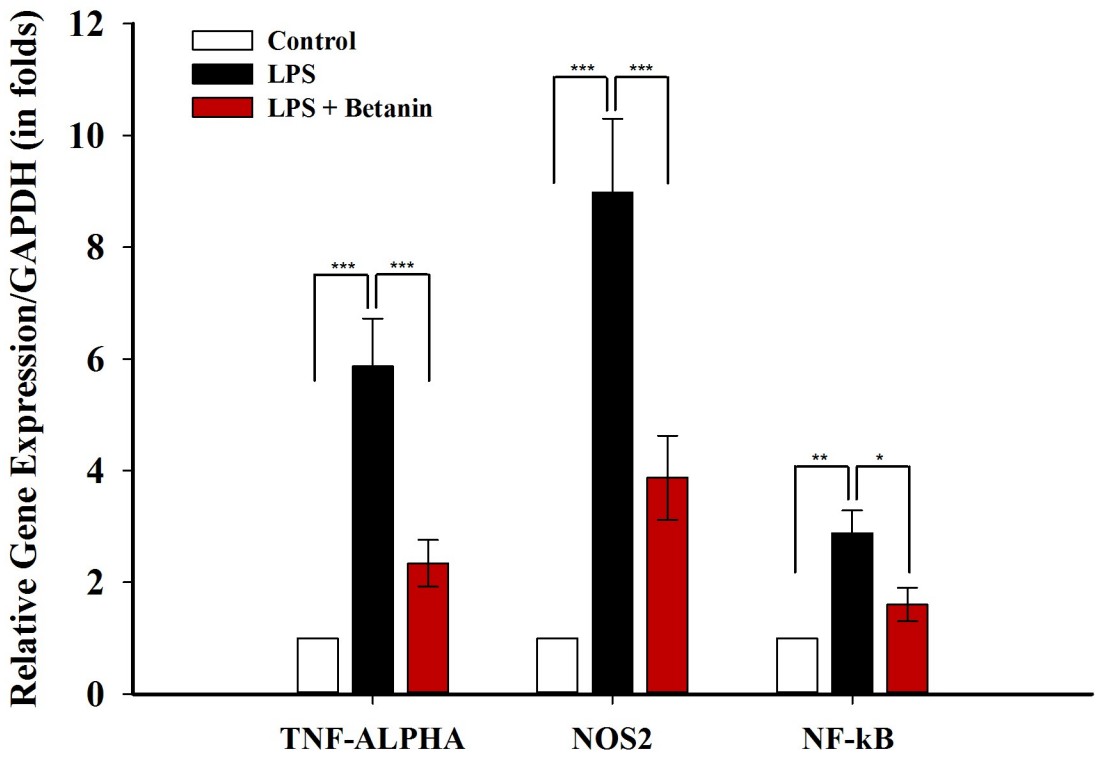

**Fig 5. Comparison of *Tnf-α*, *Nos2* and *Nf-κb* expression.** Changes in gene expression of *Tnf-α*, *Nos2* and *Nf-κb* in microglial cells under the influence of betanin pretreatment at 500 μM. *P<0.05, **P<0.01, ***P<0.001, one-way ANOVA.

cytokines such as TNF-α, IL-6, IL-1β and iNOS [52]. Hence, protein modeling for TNF-α, IL-6, IL-1β and iNOS using three servers; I-TASSER, RaptorX, and SWISS-MODEL was performed. Structure validation using SAVES v5.0 showed that SWISS-MODEL is the most appropriate method for all of our four proteins. These structures did not include any residues in the disallowed regions in the Ramachandran plot (Fig 6). In the next step, the molecular dynamic simulation was used to energy minimization of all four proteins (Fig 6). Based on the accurate docking, betanin expressed significant inhibition against TNF-α active site, with the affinity of -9.6 kcal/mol, and in this docked conformation, Val152, PRO178, PRO180, SER177, TYR193, and GLU194 were the critical amino acids that involved in hydrogen bonds. The affinity between betanin and iNOS active site was -9.1 kcal/mol with ASP289, ASP253, ARG255, PRO492, THR495, and GLN307 involved in hydrogen bonds. Also, the affinity between betanin and IL6 was -7.3 kcal/mol and CYS76, ARG66, GLU199, GLU37, and PHE35 interacted with hydrogen bonds. Eventually, the affinity of betanin with NF-κB-p65 as a critical element of neuroinflammation was -8.5 kcal/mol. In this interaction, ASN190, ARG274, SER281, LYS28 were involved. Finally, betanin displayed interactions with active site of IL6 with the affinity of -7.3 kcal/mol and CYS76, ARG66, GLU199, GLU37, and PHE35 interacted with hydrogen bonds (Fig 6). These results suggest that the anti-inflammatory properties of betanin is probably related to its potential to bind cytokines (TNF-α and IL6), NF-κB p65 and iNOS active sites. Therefor betanin can affect TNF-α, NF-κB and IL-6 in both gene expression

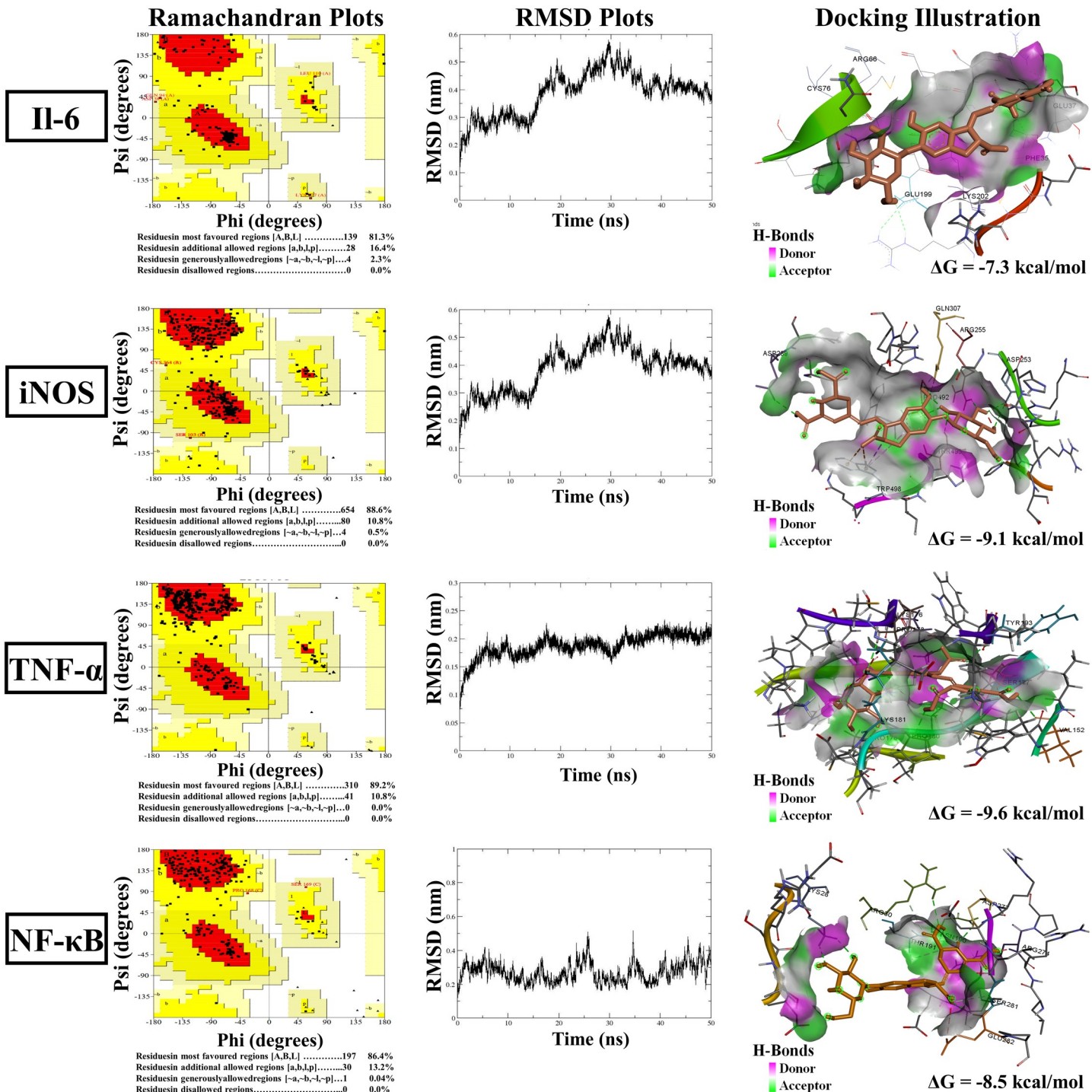

**Fig 6. Protein modeling and docking study.** Ramachandran and RMSD analysis plots of IL6, iNOS, TNF-α and NF-κB proteins modeling showed that, these structures did not contain any residues in the disallowed regions. According to the RMSD values these proteins were stable in the simulation time period. The visual illustration showed the binding energy between betanin and IL6 (ΔG = -7.3 kcal/mol), iNOS (ΔG = -9.1 kcal/mol), TNF-α (ΔG = -9.6 kcal/mol) and NF-κB (ΔG = -8.5 kcal/mol). The docked pose of betanin and each protein showed the key hydrogen-bonds area by using AutoDock Vina results. All visual illustration of the interactions generated with Discovery Studio visualizer 19.1.0.219 (https://www.3dsbiovia.com/products/collaborative-science/biovia-discovery studio/visualization-download.php) as a free resource.

and in silico level that eventually results in anti-inflammatory properties. According to the

docking results, betanin could affect TNFR1 pathway by inhibiting TNF-α and iNOS activity. Thus it can destroy the positive feedback loop between NO•, TNF-α, and iNOS.

## Conclusion

In summary, our purification methodology yielded 500±22 mg of betanin from 100 g of red beetroot. The efficiency of purification is about five times higher than the maximum value that has already been reported in other studies. The purified betanin has no toxicity on microglial cells. Therefore the anti-inflammatory effect of betanin on the activated microglial was determined. The results showed that betanin inhibited LPS-induced production of inflammatory mediators such as TNF-α, IL-1β, IL-6, free radicals, and modulates MMP, LMP, and ATP in microglial cells at 500μM concentration. Moreover, Docking results demonstrated a significant negative binding energy against active sites of TNF-α, IL-6, iNOS and NF-κB that reduce activation of LPS-induced microglial cells. These data suggest that multi-functional therapeutic applications of betanin are presumably served by the anti-inflammatory action of betanin on activated microglial.

## Author Contributions

**Formal analysis:** Hosein Ahmadi.

**Funding acquisition:** Farzaneh Sabouni.

**Investigation:** Hosein Ahmadi, Zahra Nayeri.

**Methodology:** Hosein Ahmadi, Zahra Nayeri, Zarrin Minuchehr, Mehdi Mohammadi.

**Project administration:** Mehdi Mohammadi.

**Software:** Zahra Nayeri.

**Supervision:** Farzaneh Sabouni, Mehdi Mohammadi.

**Validation:** Zarrin Minuchehr, Mehdi Mohammadi.

**Writing – original draft:** Hosein Ahmadi.

**Writing – review & editing:** Zarrin Minuchehr, Mehdi Mohammadi.

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
