## [Decision Letter · Decision Letter 0]

8 Apr 2020

PONE-D-20-07544

Betanin purification from red beetroots and evaluation of its anti-oxidant and anti-inflammatory activity on LPS-activated microglial cells

PLOS ONE

Dear Dr mohammadi,

Thank you for submitting your manuscript to PLOS ONE. After careful consideration, we feel that it has merit but does not fully meet PLOS ONE’s publication criteria as it currently stands. Therefore, we invite you to submit a revised version of the manuscript that addresses the points raised during the review process.

We would appreciate receiving your revised manuscript by May 23 2020 11:59PM. To enhance the reproducibility of your results, we recommend that if applicable you deposit your laboratory protocols in protocols.io, where a protocol can be assigned its own identifier (DOI) such that it can be cited independently in the future. For instructions see: http://journals.plos.org/plosone/s/submission-guidelines#loc-laboratory-protocols

We look forward to receiving your revised manuscript.

Kind regards,

Gautam Sethi

Academic Editor

PLOS ONE

Journal Requirements:

1. We understand that you purchased red beetroots from your local

market for this study. In your Methods section, please provide additional details regarding the source of this material. Please provide the geographic coordinates and names of the purchase location. if available, as well as any further details about the purchased items (e.g., lot number, source origin, description of appearance) to ensure reproducibility of the analyses.

Additional Editor Comments (if provided):

The authors should check the effect of Betanin on NF-kB activation as suggested by Reviewer#2.

The discussion section should be improved by discussing relevant literature such as PMID: 31315033; PMID: 25083991; PMID: 31085174; PMID: 18566231 etc.

Reviewers' comments:

Reviewer's Responses to Questions

**Comments to the Author**

1. Is the manuscript technically sound, and do the data support the conclusions?

Reviewer #1: Yes

Reviewer #2: Partly

2. Has the statistical analysis been performed appropriately and rigorously? 

Reviewer #1: Yes

Reviewer #2: Yes

3. Have the authors made all data underlying the findings in their manuscript fully available?

Reviewer #1: Yes

Reviewer #2: Yes

4. Is the manuscript presented in an intelligible fashion and written in standard English?

Reviewer #1: Yes

Reviewer #2: Yes

5. Review Comments to the Author

Reviewer #1: Mehdi and colleages have purified betanin and characterized with TLC, UV-visible, HPLC, ESI-MASS, FT-IR spectroscopy.

Investigation on the inhibitory effect of betanin on activated microglia was evaluated and authirs have displayed betanin

inhibited lipopolysaccharide induced microglial, generation of nitric oxide free radicals, reactive oxygen species, tumor

necrosis factor-alpha (TNF-α), interleukin-6 (IL-6) and interleukin-1 beta (IL-1β). Their docking results demonstrated that

betanin have moderate negative binding energy against active sites of TNF-α, IL-6 and iNOS.

I have one major issue which is with the resolution and presentation of Figure-6 which is docking figure. Author should make it more visible and easily understandable while adding appropriate leveling and also mention values for negative binding energy.

Reviewer #2: The manuscript by Ahmadi et al. entitled “Betanin purification from red beetroots and evaluation of its anti-oxidant and anti-inflammatory activity on LPS-activated microglial cells” describes the anti-oxidant and anti-inflammatory role of natural product betamin. Although the findings in this manuscript are potentially interesting, there are still some questions need to be addressed. Here is a list of several comments:

1. Figures are not well-prepared. The resolution of all the figures is very poor.

2. For FACS staining of microglial cells, combination of CD11b and CD45 labelling will be more proper otherwise it’s very hard to distinguish microglia from macrophages. Alternatively, specific microglial marker MEM119 can be used.

3. For its anti-inflammatory activity on LPS-activated microglial cells, the author should also validate by checking phospho-NF-κB p65. As another master regulator of inflammation, does p38 signalling also involve in betamin mediated anti-inflammatory activity?

6. PLOS authors have the option to publish the peer review history of their article (what does this mean?). If published, this will include your full peer review and any attached files.

Reviewer #1: No

Reviewer #2: No

---

## [Author Response · Author response to Decision Letter 0]

27 Apr 2020

Editor Comments (if provided):

1. We understand that you purchased red beetroots from your local

market for this study. In your Methods section, please provide additional details regarding the source of this material. Please provide the geographic coordinates and names of the purchase location. if available, as well as any further details about the purchased items (e.g., lot number, source origin, description of appearance) to ensure reproducibility of the analyses.

Related explanation was added to the Materials section

The authors should check the effect of Betanin on NF-kB activation as suggested by Reviewer#2.

Response:

We tried to perform more experiments as you suggested but due to the Coronavirus outbreak, the laboratories are still closed, and also having access to materials for such experiments will take 2-3 months in this new situation. However, we added more explanation from the results already published in the literature that may be a proper response for this question as below (same explanation was also added to the manuscript).

In LPS-activated macrophages, p38 immediately phosphorylated at a tyrosine residue and performs a crucial role in inflammation . Activating of p38 MAPKs is involved in the upregulation of inflammatory mediators, including TNF-α, iNOS, and cyclooxygenase-2 (COX-2) (1). On the other hand, activation of the p38 and NF-κB pathways could be accomplished by the TNFR1 and TNF-α (2). The obtained results of attenuating in the expression of Tnf-α and Nos2 at mRNA level, TNF-α protein, and the product of iNOS (NO•), could be attributed to either the direct effect of p38 on these factors or the negative effect of betanin on the performance of p38. 

The discussion section should be improved by discussing relevant literature such as PMID: 31315033; PMID: 25083991; PMID: 31085174; PMID: 18566231 etc.

This section was improved by adding more discussion accordingly.

Comments to the Author

Reviewer #1: 

I have one major issue which is with the resolution and presentation of Figure-6 which is docking figure. Author should make it more visible and easily understandable while adding appropriate leveling and also mention values for negative binding energy

Response:

Thank you for your comment. Modification on all figures including Figure 6 was made to improve their qualities.

Reviewer #2: 

1. Figures are not well-prepared. The resolution of all the figures is very poor.

Response:

The quality of all the figures were improved according your suggestion

2. For FACS staining of microglial cells, combination of CD11b and CD45 labelling will be more proper otherwise it’s very hard to distinguish microglia from macrophages. Alternatively, specific microglial marker MEM119 can be used.

Response:

We use CD11b as the most useful immunohistochemical marker for microglial cells in healthy brains (3–5). Similarly, several studies have also used cd11b alone to identify microglial cells especially in in-vitro studies like the primary culture of microglial cells (6–9). However, more specific microglial marker TMEM119 could be used to distinguish microglial cells from other immune cells (10).

3. For its anti-inflammatory activity on LPS-activated microglial cells, the author should also validate by checking phospho-NF-κB p65. As another master regulator of inflammation, does p38 signalling also involve in betamin mediated anti-inflammatory activity?

Response:

We tried to perform more experiments as you suggested but due to the Coronavirus outbreak, the laboratories are still closed, and also having access to materials for such experiments will take 2-3 months in this new situation. However, we added more explanation from the results already published in the literature that may be a proper response for this question as below (same explanation was also added to the text).

In LPS-activated macrophages, p38 immediately phosphorylated at a tyrosine residue and performs a crucial role in inflammation (11). Activating of p38 MAPKs is involved in the upregulation of inflammatory mediators, including TNF-α, iNOS, and cyclooxygenase-2 (COX-2) (1). On the other hand, activation of the p38 and NF-κB pathways could be accomplished by the TNFR1 and TNF-α (2). The obtained results of attenuating in the expression of Tnf-α and Nos2 at mRNA level, TNF-α protein, and the product of iNOS (NO•), could be attributed to either the direct effect of p38 on these factors or the negative effect of betanin on the performance of p38. 

1. Yi YS, Son YJ, Ryou C, Sung GH, Kim JH, Cho JY. Functional roles of Syk in macrophage-mediated inflammatory responses. Mediators Inflamm. 2014;2014. 

2. Blaser H, Dostert C, Mak TW, Brenner D. TNF and ROS Crosstalk in Inflammation. Trends Cell Biol. 2016 Apr;26(4):249–61. 

3. Hickstein DD, Ozols J, Williams SA, Baenziger JU, Locksley RM, Roth GJ. Isolation and characterization of the receptor on human neutrophils that mediates cellular adherence. J Biol Chem. 1987;262(12):5576–80. 

4. Sasaki A. Microglia and brain macrophages: An update. Neuropathology. 2017;37(5):452–64. 

5. Jeong H-K, Ji K, Min K, Joe E-H. Brain Inflammation and Microglia: Facts and Misconceptions. Exp Neurobiol. 2013;22(2):59–67. 

6. Lacagnina MJ, Fabisiak TJ, Grace PM. Mapping microglial reactivity in the brain after sciatic nerve injury. Brain Behav Immun [Internet]. 2017;66(2017):e16. Available from: https://doi.org/10.1016/j.bbi.2017.07.068

7. Morrison HW, Filosa JA. Sex differences in astrocyte and microglia responses immediately following middle cerebral artery occlusion in adult mice. Neuroscience [Internet]. 2016;339:85–99. Available from: http://dx.doi.org/10.1016/j.neuroscience.2016.09.047

8. Hutter-Schmid B, Humpel C. Primary mouse brain pericytes isolated from transgenic Alzheimer mice spontaneously differentiate into a CD11b+ microglial-like cell type in vitro. Exp Gerontol [Internet]. 2018;112(July):30–7. Available from: https://doi.org/10.1016/j.exger.2018.08.003

9. Jia X, Cong B, Wang S, Dong L, Ma C, Li Y. Secondary damage caused by CD11b+ microglia following diffuse axonal injury in rats. J Trauma Acute Care Surg. 2012;73(5):1168–74. 

10. Bennett ML, Bennett FC, Liddelow SA, Ajami B, Zamanian JL, Fernhoff NB, et al. New tools for studying microglia in the mouse and human CNS. Proc Natl Acad Sci U S A. 2016;113(12):E1738–46. 

11. Lee JC, Laydon JT, McDonnell PC, Gallagher TF, Kumar S, Green D, et al. A protein kinase involved in the regulation of inflammatory cytokine biosynthesis. Vol. 372, Nature. 1994. p. 739–46.

---

## [Editor Report · Decision Letter 1]

29 Apr 2020

Betanin purification from red beetroots and evaluation of its anti-oxidant and anti-inflammatory activity on LPS-activated microglial cells

PONE-D-20-07544R1

Dear Dr. mohammadi,

We are pleased to inform you that your manuscript has been judged scientifically suitable for publication and will be formally accepted for publication once it complies with all outstanding technical requirements.

With kind regards,

Gautam Sethi

Academic Editor

PLOS ONE
---

## [Editor Report · Acceptance letter]

1 May 2020

PONE-D-20-07544R1 

Betanin purification from red beetroots and evaluation of its anti-oxidant and anti-inflammatory activity on LPS-activated microglial cells 

Dear Dr. Mohammadi:

I am pleased to inform you that your manuscript has been deemed suitable for publication in PLOS ONE. Congratulations! Your manuscript is now with our production department. 

With kind regards,

on behalf of

Dr. Gautam Sethi 

Academic Editor

PLOS ONE